# In Vitro and In Vivo Pharmacological Profiles of LENART01, a Dermorphin–Ranatensin Hybrid Peptide

**DOI:** 10.3390/ijms25074007

**Published:** 2024-04-03

**Authors:** Nadine Hochrainer, Pawel Serafin, Sara D’Ingiullo, Adriano Mollica, Sebastian Granica, Marek Brytan, Patrycja Kleczkowska, Mariana Spetea

**Affiliations:** 1Department of Pharmaceutical Chemistry, Institute of Pharmacy and Center for Molecular Biosciences Innsbruck (CMBI), University of Innsbruck, 6020 Innsbruck, Austria; 2Military Institute of Hygiene and Epidemiology, 01-163 Warsaw, Poland; pawelserafin1@wp.pl (P.S.); marek.brytan@wihe.pl (M.B.); 3Department of Pharmacy, University “G. d’Annunzio” of Chieti-Pescara, Via dei Vestini 31, 66100 Chieti, Italy; sara.dingiullo001@phd.unich.it (S.D.); a.mollica@unich.it (A.M.); 4Microbiota Lab, Department of Pharmacognosy and Molecular Basis of Phytotherapy, Medical University of Warsaw, 02-097 Warsaw, Poland; sebastian.granica@wum.edu.pl; 5Maria Sklodowska-Curie Medical Academy in Warsaw, 03-411 Warsaw, Poland

**Keywords:** multitarget ligands, opioid receptors, dopamine receptors, dermorphin, ranatensin, antinociception, tail-flick test, formalin test, inflammatory pain, side effects

## Abstract

Diverse chemical and pharmacological strategies are currently being explored to minimize the unwanted side effects of currently used opioid analgesics while achieving effective pain relief. The use of multitarget ligands with activity at more than one receptor represents a promising therapeutic approach. We recently reported a bifunctional peptide-based hybrid LENART01 combining dermorphin and ranatensin pharmacophores, which displays activity to the mu-opioid receptor (MOR) and dopamine D2 receptor (D2R) in rat brains and spinal cords. In this study, we investigated the in vitro binding and functional activities to the human MOR and the in vivo pharmacology of LENART01 in mice after subcutaneous administration. In vitro binding assays showed LENART01 to bind and be selective to the human MOR over the other opioid receptor subtypes and delta, kappa and nociceptin receptors. In the [^35^S]GTPγS binding assay, LENART01 acted as a potent and full agonist to the human MOR. In mice, LENART01 produced dose-dependent antinociceptive effects in formalin-induced inflammatory pain, with increased potency than morphine. Antinociceptive effects were reversed by naloxone, indicating MOR activation in vivo. Behavioral studies also demonstrated LENART01’s properties to induce less adverse effects without locomotor dysfunction and withdrawal syndrome compared to conventional opioid analgesics, such as morphine. LENART01 is the first peptide-based MOR-D2R ligand known to date and the first dual MOR-dopamine D2R ligand for which in vivo pharmacology is reported with antinociceptive efficacy and reduced opioid-related side effects. Our current findings may pave the way to new pain therapeutics with limited side effects in acute and chronic use.

## 1. Introduction

Pain is an unsolved medical condition, and it is among the most prevalent and debilitating human illnesses [1,2]. Pain is not only a disabling symptom of many medical conditions, but also a disease state in its own right. Central goals in pain control are to provide analgesia of adequate efficacy and to reduce the complications of currently available drugs. Opioid analgesics have been on the frontline of pain management for many years. However, available opioid pain medications cause serious side effects, including a high burden of abuse and addiction [3,4,5]. The rapid increase in the use of opioid drugs in the United States has been termed the “opioid crisis”, with more than 80,000 opioid-related deaths reported in 2021 [6].

The analgesic effects of opioids are mediated through four receptors, namely mu-(MOR), kappa-(KOR), delta-(DOR) and nociceptin/orphanin FQ (NOP) [5,7]. All four opioid receptors are members of the seven-transmembrane-spanning G protein-coupled receptor (GPCR) family and are expressed throughout the central and peripheral nervous systems (CNS and PNS, respectively) and non-neuronal tissues [5,7]. The MOR, which is the main target of clinically used opioid medications, is central for analgesia, but it is also responsible for undesirable side effects, particularly addiction and abuse liabilities [5,7,8].

Because of the inadequate benefit/risk ratio of currently available opioids, together with the ongoing opioid epidemic, there is a huge quest for strategies to discover new opioid analgesics. Alternative chemical and pharmacological approaches are therefore explored to mitigate the deleterious effects of MOR agonists and to limit their abuse and misuse [9,10,11,12,13,14,15,16,17,18,19], amongst which are multifunctional drugs. The concept of ‘one molecule, multiple targets’ has received increased attention in opioid drug discovery as a promising strategy to generate new analgesics with enhanced effectiveness and reduced unwanted side effects. Accumulated preclinical data were reported on multifunctional analgesics targeting opioid/opioid and opioid/non-opioid receptors, including both small molecules and peptide-based ligands [20,21,22,23,24,25,26,27]. Furthermore, the application of two or more biologically active components in the form of a single molecule can significantly improve the physicochemical and/or pharmacokinetic properties of CNS activity.

Whereas opioid analgesics and other addictive substances (e.g., cocaine and cannabinoids) act directly on their specific receptors, they also indirectly influence the dopamine system that controls reward behavior [5,28,29,30]. The reward system consists of dopaminergic neurons, whose activation causes an increase in dopamine release in mesolimbic structures, resulting in a rewarding effect. Among the dopamine receptor subtypes, the D1, D2 and D3 receptors (D1R, D2R and D3R, respectively) have been reported to play important roles in the reinforcement of drug-seeking behavior [31,32,33,34,35,36,37]. Recent data suggest using dopamine D1R or D3R preferring modulators to prevent morphine tolerance and to reduce the duration of morphine withdrawal symptoms [38].

Thus, in light of recent research, bifunctional ligands encompassing pharmacophores targeting both opioid and dopamine receptors received interest as potentially safer analgesics. Designing dual opioid–dopamine receptor ligands may be an attractive way to balance the unwanted side effects derived from their individual components. The first described bifunctional opioid–dopamine receptor ligands were small molecules with a dual-target binding on the MOR and dopamine D3R, with a MOR agonism and D3R antagonism/partial agonism [39,40]. Peptides and peptide analogues offer several advantages over small molecules in various biomedical applications, including pain [41,42,43,44,45]. Some benefits of peptides include high target specificity, enhanced potency, improved selectivity, reduced off-target engagement, reduced toxicity and modularity for future pharmacokinetic optimization and formulation.

We recently reported on a bifunctional peptide-based hybrid LENART01, which combines an MOR pharmacophore and a dopamine D2R pharmacophore [46]. Our design strategy for LENART01 was based on merging dermorphin and ranatensin by following the principle of pharmacophore fusion (Figure 1). The N-terminal pharmacophore of LENART01 consisted of dermorphin with an amino acid sequence of YdAFGYPS, and the C-terminus was constructed from ranatensin, which was modified, resulting in a sequence of GHFM. The opioid pharmacophore dermorphin is a heptapeptide isolated from the skin of the Amazon frog *Phyllomedusa sauvagei* [47]. Dermorphin is a selective MOR agonist and potent antinociceptive used as a lead for the design of dermorphin analogues [48,49,50,51]. In early clinical studies, dermorphin administered via the intrathecal (i.t.) route had increased effectiveness than morphine in treating postoperative pain [52]. Ranatensin is an undecapeptide that is first isolated from the skin of the frog *Rana pipiens* [53,54]. Ranatensin belongs to the bombesin-like peptide family and shares significant homology with bombesin [55,56]. In vitro and in vivo studies described ranatensin to have activity at the dopamine D2R and to attenuate pain behavior in mice after central intracerebroventricular (i.c.v.) administration, with dopamine neurotransmission contributing to the antinociceptive action of ranatensin [57,58].

In vitro binding studies established LENART01 to bind to and activate both MOR and D2R in rat brains and spinal cords. Furthermore, it showed high selectivity to the D2R compared to D1R [46]. We also described LENART01’s antimicrobial activity against different *E. coli* strains [59]. With the crystal structures of the active human MOR and available dopamine D2R, an initial in silico evaluation using molecular docking and molecular dynamics (MD) simulations was conducted on the binding mode and interaction mechanisms of LENART01 with the two GPCRs [46].

Because of the interesting profile of LENART01, further investigations into its in vitro and in vivo pharmacology were warranted. In this study, we characterized LENART01 for an in vitro profile to human opioid receptors (binding, selectivity and agonist activities). The in vivo behavioral properties were evaluated in mouse models of acute and inflammatory pain after subcutaneous (s.c.) administration, together with the potential for inducing opioid liabilities of locomotor dysfunction and withdrawal response. Studies on the mechanism underlying the antinociceptive effect of LENART01 were also undertaken.

## 2. Results and Discussion

### 2.1. LENART01 Displays Selectivity and Full Agonist Activity In Vitro to Human MOR

We previously reported on the specific binding of LENART01 to the rat MOR and rat dopamine D2R in the brain and spinal cord membrane preparations and its high selectivity to the dopamine D2R compared to D1R [46]. Because the human MOR is the ultimate target of therapeutic opioid drugs, we evaluated LENART01’s binding properties to the human MOR through in vitro radioligand competition binding to membrane preparations from Chinese hamster ovary (CHO) cells stably expressing the recombinant MOR as described [60]. Moreover, LENART01’s receptor binding/selectivity to the other opioid receptor subtypes, the DOR, KOR and NOP receptors, was not reported. Consequently, we examined LENART01’s binding profile to the human DOR, KOR and NOP receptors expressed in CHO cells. The opioid binding profile of LENART01 was compared to that of dermorphin.

As shown in Figure 2, LENART01 and dermorphin produced a concentration-dependent inhibition of the selective opioid radioligands, [^3^H]DAMGO, [^3^H]DPDPE and [^3^H]U69,593, binding from the human MOR, DOR and KOR, respectively. Binding studies of the human NOP receptor using [^3^H]nociceptin indicated no substantial binding of LENAR01 and dermorphin at the concentration of 10 µM (Figure 2D). The binding affinities were calculated and are presented in Table 1. We established that LENART01 bound to the human MOR with high affinity (K_i_ = 11 nM) and reduced binding to the human DOR and KOR based on the higher K_i_ values to these two receptors (Table 1). In competition binding assays, dermorphin also showed high binding affinity to the MOR (K_i_ = 1.89 nM) and selectivity for the MOR, which aligned with previously reported data on the neuronal MOR in rat brains [61,62] or to the recombinant MOR expressed in CHO or human embryonic kidney (HEK293) cells [63,64]. When compared to the binding profile of dermorphin, LENART01 displayed a decreased affinity to the human MOR by around six-fold, whereas a maximum of a two-fold decrease was calculated for binding to the human DOR and KOR (Table 1). Notably, LENART01 maintained the MOR selectivity of dermorphin, though with lower selectivity ratios for MOR versus the other opioid receptor subtypes.

Initial in vitro data on the opioid functional activity of LENART01 were reported in the rat brain and spinal cord preparations using the guanosine-5′-*O*-(3-[^35^S]thio)triphosphate ([^35^S]GTPγS) binding assay [46]. LENART01 was depicted as an agonist to the rat MOR. In this study, we evaluated the agonist activity of LENART01 to the human MOR using the [^35^S]GTPgS binding assay with CHO-hMOR cell membranes as described previously [60]. Because LENART01 showed reduced to no specific binding to the other human opioid receptor types (DOR, KOR and NOP receptors) (Table 1), the [^35^S]GTPgS binding assay was not carried out for these receptors. The agonist profile of LENART01 to the human MOR was compared to that of dermorphin (Figure 3). The potencies are shown as EC_50_ values and the efficacies are shown as % stimulation to the reference MOR agonist DAMGO in Table 2.

As shown in Figure 3, LENART01 increased [^35^S]GTPγS binding to the human MOR in a concentration-dependent manner, demonstrating full efficacy compared to DAMGO and dermorphin. In this functional assay, LENAR01 displayed potent MOR agonist activity, with an ED_50_ value only two-fold compared to DAMGO, and a five-fold reduced potency compared to dermorphin (Table 2). The in vitro agonist profile of dermorphin to the human MOR established in our study is in good agreement with the functional data reported earlier [63,64].

### 2.2. LENART01 Shows Antinociceptive Efficacy with Opioid-Mediated Action in Formalin Test after Subcutaneous Administration in Mice

With the knowledge that both peptides, dermorphin and ranatensin, the two pharmacophores comprising LENART01, have antinociceptive effects [48,57], we next evaluated the in vivo pharmacology of LENART01 in terms of antinociceptive efficacy. First, the behavioral properties of LENART01 were assessed in a mouse model of acute nociceptive pain using the radiant heat tail-flick test [65] after s.c. administration in mice. The tail withdrawal latencies of mice to thermal stimulation were measured as described previously [66]. As shown in Figure 4A, the s.c. administration of LENART01 to mice did not produce a significant increase in the tail-flick latencies at any time point and tested doses (7.8 and 39 µmol/kg). In the same pain assay, conventional opioids including morphine, oxycodone, oxymorphone, buprenorphine and fentanyl were well-established to induce potent antinociceptive effects following systemic administration to rodents [66,67,68,69]. Activity in the tail-flick test suggests that a drug acts via CNS and may reflect spinal activity [70].

To further investigate the therapeutic antinociceptive potential of LENART01, we examined its in vivo actions in a model of inflammatory pain using the formalin test [71]. Pain behavior was assessed in mice that received an s.c. injection of formalin solution to the plantar surface of the right hind paw as described previously [60].

The systemic s.c. administration of LENAR01 produced time- and dose-dependent reductions in pain behavior in the formalin-injected mice, determined as the amount of time each animal spent licking, biting, lifting and flinching the formalin-injected paw (Figure 4B). LENART01 attenuated the pain behavior during the nociceptive Phase I and inflammatory Phase II of the formalin assay with a significant effect at all tested doses (Figure 4C,D). The inhibition in pain behavior expressed as % of antinociception during Phase I and Phase II is presented in Table 3. The calculated antinociceptive effective dose ED_50_ value of LENART01 in the inflammatory Phase II of the formalin test was 1.60 µmol/kg. We recently reported on the antinociceptive efficacy of morphine in the formalin test in mice after s.c. administration, with morphine having an ED_50_ value of 6.44 µmol/kg [60]. When comparing the antinociceptive potency of LENART01 to morphine, it is evident that LENART01 shows antinociceptive potency around 4-fold higher than morphine in mice with inflammatory pain.

To evaluate the mechanism of action, i.e., the involvement of the opioid receptors, in the antinociceptive effect of LENART01 in the formalin test, the effect of the opioid antagonist naloxone was tested (Figure 4E,F). Pre-treating mice with naloxone (1 mg/kg, s.c.) resulted in a significant and complete reversal of the antinociceptive response of LENART01 in both Phases I and II, signifying that opioid receptors, specifically the MOR, given the MOR selectivity (Table 1), are involved in LENART01’s in vivo agonist activity.

### 2.3. LENART01 Does Not Cause Sedation or Affect Motor Coordination after Subcutaneous Administration in Mice

Conventional opioid analgesics, such as morphine, oxycodone, oxymorphone, buprenorphine and fentanyl, are known to produce sedation and alter locomotor activity, which represent undesirable side effects that limit their clinical usefulness [5,66,67,68]. To further address the behavioral properties of LENATR01, we investigated its effects on motor coordination and its potential to induce sedation after s.c. administration in mice using the rotarod test, a well-established model for evaluating the loss of coordinated locomotion [72]. The latencies of mice to fall off the rotarod were assessed as described previously [66].

The dose-dependent drug effects on motor performance induced by LENATR01 in the mouse rotarod test are shown in Figure 5. Mice were administered LENATR01 at doses corresponding to 5- and 24-fold the antinociceptive ED_50_ dose in the formalin test (Table 3). LENART01 did not produce changes in the motor behaviors of mice, as no significant alterations in the rotarod latencies were measured at any time point and tested dose (7.8 and 39 µmol/kg) compared to the saline-treated animals. In the same test, we reported that mice that were s.c. treated with morphine at a dose of 31 µmol/kg (i.e., 10 mg/kg) showed a significantly reduced time on the rotarod [66,68]. Based on these data, we show the safe profile of LENART01 regarding sedation and locomotor activity following s.c. administration in mice.

### 2.4. Chronic Administration of LENART01 Does Not Induce Withdrawal Syndrome in Mice

The long-term use of traditional opioids is associated with physical dependence [5]. Such dependence can be visualized in opioid-dependent mice by injecting naloxone and the occurrence of withdrawal symptoms, i.e., jumps, paw tremors, head shakes and diarrhea [73,74]. To evaluate the behavioral effects of LENAR01 in mice after chronic s.c. treatment, the potential for physical dependence was determined using naloxone-precipitated withdrawal syndrome as described previously [60]. Mice were treated twice daily over a 5-day period with a dose of 7.8 µmol/kg of LENAR01 or saline (control). Mice were administered LENART01 in a dose showed to be effective in the formalin test, corresponding to 5-fold the antinociceptive ED_50_ dose (Table 3).

Administration of naloxone (1 mg/kg, s.c.) two hours after the last s.c. injection of LENART01 did not induce withdrawal signs when compared to the saline-treated mice (Figure 6). In contrast, we recently reported on significant naloxone-induced withdrawal syndrome following repeated s.c. administration of morphine in a dose of 15 µmol/kg (5 mg/kg) to mice [60]. Furthermore, the chronic administration of 7.8 µmol/kg of LENART01 did not have any significant effect on the body weights of mice compared to the saline-treated animals (Figure 7). The morphine-dependent animals showed decreased dopamine release in the nucleus accumbens during withdrawal [75,76,77]. The chronic administration of morphine causes a decrease in the dopamine D2 receptor mRNA expression in the brain [78]. Our behavioral data indicate that the lack of LENART01 to induce withdrawal syndrome may be associated with its ability to affect D2R gene expression and due to its D2R agonist activity.

Our behavioral data show that LENAR01 produces significant and potent antinociception in a mouse model of acute inflammatory pain without the opioid-mediated liability of physical dependence after repeated s.c. administration in mice. 

### 2.5. LENART01 Shows a Reduced Capability to Enter the Brain

We next evaluated the in vivo pharmacokinetic profile of LENART01 after s.c. administration in mice. Mice received 7.8 and 39 µmol/kg, and brains were dissected two hours after the drug treatment and processed for analysis. Dermorphin is known to have low CNS permeability and bioavailability [79], whereas no information is available on ranatensin. However, it is assumed that ranatensin is also deprived of the ability to penetrate the blood–brain barrier (BBB), as the permeability of the BBB to bombesin is reduced [80]. Hybrid compounds are known to possess new properties, which may include transport through the BBB, despite their molecular weight of more than 500 Daltons. However, the hybridization of dermorphin- and ranatensin-based pharmacophores in LENART01 resulted in a complete lack of transport across the BBB, as the HPLC chromatograms showed no detectable signal (Figure 8). This finding is also supported by behavioral observations as LENART01 did not elicit a spinal nociceptive reflex in the radiant heat tail-flick test (Figure 4A), and there was a lack of sedation/motor incoordination (Figure 5) after systemic s.c. administration in mice.

## 3. Materials and Methods

### 3.1. Drugs and Chemicals

LENART01 was prepared as described previously [59]. Radioligands [^3^H]DAMGO (51.7 Ci/mmol), [^3^H]DPDPE (47.4 Ci/mmol), [^3^H]U69,593 (49.3 Ci/mmol), [^3^H]nociceptin (119.4 Ci/mmol) and [^35^S]GTPγS (1250 Ci/mmol) were purchased from PerkinElmer (Boston, MA, USA). Guanosine diphosphate (GDP), GTPγS, DAMGO, DPDPE, U69,593, nociceptin, formalin, polyethylenimine (PEI), tris(hydroxymethyl) aminomethane (Tris), 2-[4-(2-hydroxyethyl)piperazin-1-yl]ethanesulfonic acid (HEPES), bovine serum albumin (BSA) and cell culture media and supplements were obtained from Sigma-Aldrich Chemicals (St. Louis, MO, USA). Dermorphin was obtained from MCE (Sollentuna, Sweden). Naloxone hydrochloride was kindly provided by Dr. Helmut Schmidhammer (University of Innsbruck, Innsbruck, Austria). All other chemicals were of analytical grade and obtained from standard commercial sources. Test compounds were prepared as 1 mM stocks in water for in vitro assays or dissolved in 0.9% physiological saline for in vivo assays and further diluted to working concentrations in the appropriate medium.

### 3.2. Cell Culture and Membrane Preparation

CHO cells stably expressing the human opioid receptors (CHO-hMOR, CHO-hDOR, CHO-hKOR and CHO-hNOP) were kindly provided by Dr. Lawrence Toll (SRI International, Menlo Park, CA, USA). CHO-hMOR and CHO-hDOR cells were cultured in Dulbecco’s Modified Eagle’s Medium (DMEM)/Ham’s F12 culture medium supplemented with 10% fetal bovine serum (FBS), 0.1% penicillin/streptomycin, 2 mM L-glutamine and 0.4 mg/mL geneticin (G418). CHO-hKOR and CHO-hNOP cells were cultured in DMEM culture medium supplemented with 10% FBS, 0.1% penicillin/streptomycin, 2 mM L-glutamine and 0.4 mg/mL geneticin (G418). All cell cultures were grown at 37 °C in a humidified atmosphere of 95% air and 5% CO_2_. Membranes from CHO-hOR cells were prepared as previously described [60]. Briefly, cells grown at confluence were removed from the culture plates by scraping, homogenized in 50 mM Tris-HCl buffer (pH 7.7) using a Dounce glass homogenizer, and then centrifuged once and washed by additional centrifugation at 27,000× *g* for 15 min at 4 °C. The final pellet was resuspended in 50 mM Tris-HCl buffer (pH 7.7) and stored at −80 °C until use. Protein content of cell membrane preparations was determined by the method of Bradford using BSA as the standard [81].

### 3.3. Radioligand Competitive Binding Assays for Human Opioid Receptors

Competitive binding assays were conducted on human opioid receptors stably transfected into CHO cells according to the published procedures [60]. Binding assays were performed using [^3^H]DAMGO (1 nM), [^3^H]DPDPE (1 nM), [^3^H]U69,593 (1 nM) or [^3^H]nociceptin (0.1 nM) for labeling MOR, DOR, KOR or NOP receptors, respectively. Non-specific binding was determined using 10 µM of the unlabeled counterpart of each radioligand assays were performed in 50 mM Tris-HCl buffer (pH 7.4) in a final volume of 1 mL. In the NOP receptor binding assay, 1 mg/mL BSA was added to the assay buffer. Cell membranes (15–20 µg) were incubated with various concentrations of test compounds and the appropriate radioligand for 60 min at 25 °C. After incubation, reactions were terminated by rapid filtration through Whatman GF/C glass fiber filters. In the NOP receptor binding assay, filtration was made through 0.5% PEI-soaked Whatman GF/B glass fiber filters. Filters were washed three times with 5 mL of ice-cold 50 mM Tris-HCl buffer (pH 7.4) using a Brandel M24R cell harvester (Gaithersburg, MD, USA). Radioactivity retained on the filters was counted by liquid scintillation counting using a Beckman Coulter LS6500 (Beckman Coulter Inc., Fullerton, CA, USA). All experiments were performed in duplicate and repeated three times with independently prepared samples.

### 3.4. [^35^S]GTPγS Binding Assay for Human MOR

The binding of [^35^S]GTPγS to membranes from CHO stably expressing the human MOR (CHO-hMOR) was conducted according to the procedure published in [60]. Cell membranes (10–15 µg) in Buffer A (20 mM HEPES, 10 mM MgCl_2_ and 100 mM NaCl, pH 7.4) were incubated with 0.05 nM [^35^S]GTPγS, 10 µM GDP and various concentrations of test compounds in a final volume of 1 mL for 60 min at 25 °C. Non-specific binding was determined using 10 µM GTPγS, and the basal binding was determined in the absence of the test compound. Samples were filtered over Whatman GF/B glass fiber filters using a Brandel M24R cell harvester (Brandel, Gaithersburg, MD, USA). Radioactivity retained on the filters was counted by liquid scintillation counting using a Beckman Coulter LS6500 (Beckman Coulter Inc., Fullerton, CA, USA). All experiments were performed in duplicate and repeated three times with independently prepared samples.

### 3.5. Animals and Drug Administration

Experiments were performed with male CD1 mice (8–10 weeks old, 30–35 g body weight) purchased from Janvier Labs (Le Genest-Saint-Isle, France). All animal care and experimental procedures were in accordance with the ethical guidelines for the animal welfare standards of the European Communities Council Directive (2010/63/EU) and were approved by the Committee of Animal Care of the Austrian Federal Ministry of Science and Research. Mice were group-housed (maximum of 5 animals per cage) in a temperature- (21–22 °C) and humidity-controlled (60–70%) specific pathogen-free room with a 12 h light/dark cycle and with free access to food and water. LENART01 and naloxone were prepared in sterile physiological 0.9% saline solution. Tested doses of LENART01 were 0.78, 3.9, 7.8 and 39 µmol/kg (1, 5, 10 and 50 mg/kg, respectively). LENART01 or vehicle (saline) were s.c. administered in a volume of 10 µL/g body weight.

### 3.6. Tail-Flick Assay

The radiant heat tail-flick test was performed using a UB 37360 Ugo Basile analgesiometer (Ugo Basile s.r.l., Varese, Italy) as described previously [66]. The reaction time required by the mouse to remove its tail after the application of radiant heat was measured and defined as the tail-flick latency (in seconds). Tail-flick latencies were measured before and after the s.c. administration of saline or LENART01 (i.e., 15, 30, 60 and 120 min) (test latency, TL). A cut-off time of 10 s was used in order to minimize tissue damage. The antinociceptive effect (as percentage of maximum possible effect, % MPE) was calculated according to a formula ([(TL − BL)/(cut-off time − BL)] × 100), where TL represents the test latency and BL is the basal latency. Each experimental group included 5 mice.

### 3.7. Formalin Test

The formalin-induced inflammatory pain test was performed as described previously [60]. Following a habituation period of 15 min to individual transparent observation chambers, the mice were s.c. administered saline or different doses of the LENART01 15 min prior to the injection of 20 µL of 5% formalin aqueous solution to the plantar surface of the right hind paw. Each mouse was observed for 40 min in 5 min intervals following the injection of formalin. The amount of time (in seconds, sec) each animal spent licking, biting, lifting and flinching the formalin-injected paw (pain behavior) was recorded during Phase I (0–5 min) and Phase II (15–40 min). In the antagonism study, naloxone (1 mg/kg) was s.c. injected 15 min before the s.c. administration of LENART01 (7.8 mg/kg), and pain behavior was assessed as described above. Antinociceptive effect expressed in percentage (%) was calculated according to the following formula: 100 × [(C − T)/C]. C is the mean time in the control (saline) group and T is the time in the drug-treated group. The dose–response relationships of the percentage inhibition of pain behavior were constructed, and the dose necessary to produce a 50% effect (ED_50_) and 95% confidence limits (95% CL) was calculated according to the method of Litchfield and Wilcoxon [82]. Each experimental group included 6–8 mice.

### 3.8. Rotarod Test

The rotarod test was performed using an accelerating rotarod treadmill (Acceler Rota-Rod 7650, Ugo Basile s.r.l., Varese, Italy) for mice (diameter 3.5 cm) as described previously [66]. Animals were accustomed to the equipment in two training sessions (30 min apart) one day before testing. On the experimental day, mice were placed on the rotarod, and treadmill was accelerated from 4 to 40 rpm over a period of 5 min. The time spent on the drum was recorded for each mouse before (baseline) and after the s.c. administration of saline or test compound (i.e., 30, 60 and 120 min). Decreased latencies to fall in the rotarod test indicate impaired motor performance. A 300 sec cut-off time was used. The rotarod data are expressed as percentage (%) changes from the rotarod latencies obtained before (baseline, B) and after drug administration (test, T) and were calculated as follows: 100 × (T/B). Each experimental group included 5 mice.

### 3.9. Naloxone-Precipitated Withdrawal Syndrome

Opioid physical dependence was assessed using naloxone-induced withdrawal syndrome in mice as described previously [60]. Mice received two s.c. injections of LENART01 (7.8. mg/kg) or saline daily over a 5-day period. On day 5, two hours after the last drug injection, the withdrawal syndrome was precipitated through the administration of naloxone (1 mg/kg, s.c.). Animals were immediately placed in clear acrylic cylinders, and signs of opioid withdrawal were recorded for 15 min. Monitored behaviors included vertical jumps, paw tremor, head shakes, urine output, the presence/absence of diarrhea, feces output and body weight loss. For each mouse, a global withdrawal score was calculated by summing the values obtained for each sign (one point was assigned to every 3 jumps and 5 paw tremors, respectively, whereas all other signs were given the absolute values recorded during the test). Changes in body weight were calculated as percentage (%) from the body weight on day 1 (B, before drug administration) and on each treatment day (T) as 100 × [(T − B)/B]. Each experimental group included 5–7 mice.

### 3.10. Pharmacokinetic Study

Two hours after the s.c. administration of LENART01 (doses of 7.8 and 39 µmol/kg), the mice were sacrificed and the brains were removed, washed with sterile physiological 0.9% saline and frozen in dry ice until use. For pharmacokinetic studies, brain tissue was homogenized in acetonitrile (ACN; 1:3, *w*/*v*) and centrifuged at 1800 rpm at 4 °C to collect the supernatant. The LC-DAD-MS method was used for the detection of LENART01 in the prepared samples. The analysis was performed on a UHPLC-3000RS system (Dionex, Leipzig, Germany) equipped with a DAD detector, degasser and an autosampler. The eluate was introduced without splitting to an AmaZon SL ion trap mass spectrometer with an ESI interface (Bruker Daltonik GmbH, Bremen, Germany). The MS spectra were recorded in the negative ion mode only. The UV spectra were recorded over the range of 200–600 nm. The parameters of the MS unit were as follows: nebulizer pressure of 40 psi, drying gas flow rate of 9 L/min, nitrogen gas temperature of 134 °C and capillary voltage of 4.5 kV. The mass spectra were registered by scanning from *m*/*z* 70 to 2200. The separation was carried out with Kinetex XB C_18_ column (Phenomenex, Torrance, CA, USA), 150 mm × 2.1 mm × 1.7 µm. The mobile phase (A) was H_2_O/HCOOH (100:0.1, *v*/*v*), and the mobile phase (B) was MeCN/HCOOH (100:0.1, *v*/*v*). The gradient elution was used as follows: the flow rate was 0.3 mL/min, the elution started at 0 min = 5%B and the B concentration was increased to 50%B at 13 min and finally to 65%B at 20 min. The column temperature was maintained at 25 °C. The presence of the analyzed compound was checked by the monitoring of its pseudomolecular ion eluting at ca. 10 min. For better clarity, EICs for *m*/*z* = 1274.2 ± 0.5 were created.

### 3.11. Data and Statistical Analysis

Experimental data were graphically processed and statistically analyzed using the GraphPad Prism 9.0 Software (GraphPad Prism Software Inc., San Diego, CA, USA). In in vitro assays, inhibition constant (K_i_, nM), potency (EC_50_, nM) and efficacy (E_max_, %) values were determined from concentration–response curves by nonlinear regression analysis. The K_i_ values were determined by the method of Cheng and Prusoff [83]. In the [^35^S]GTPγS binding assays, efficacy was determined relative to the reference MOR full agonist, DAMGO. For in vivo behavioral data, two-sample comparison was performed using unpaired *t*-test. For multiple comparisons between the treatment groups, ANOVA (one-way or two-way, as appropriate) with Tukey’s or Bonferroni’s *post hoc* tests were used. All data are presented as mean ± SEM. A *p* < 0.05 was considered statistically significant.

## 4. Conclusions

Bifunctional ligands that simultaneously target opioid receptors along with other neurotransmitter systems involved in pain modulation and opioid-induced side effects currently hold substantial scientific and clinical interest. In this study, we reported the in vitro and in vivo pharmacology of the first dual MOR–dopamine D2R hybrid peptide, LENART01, which was designed by combining dermorphin and ranatensin pharmacophores. The in vitro binding studies showed LENART01 to display selectivity and full agonist activity to the human MOR. In the in vivo study, LENART01 produced potent antinociceptive effects in a model of inflammatory pain after s.c. administration in mice. Receptor antagonist in vivo studies established LENART01-induced antinociception to be mediated via MOR activation. Furthermore, this hybrid peptide exhibited higher antinociception potency compared to morphine and reduced MOR-mediated liabilities of physical dependence and sedation/motor dysfunction, thus demonstrating a better tolerability profile. Notably, LENART01 is the first dual MOR–dopamine D2R ligand investigated in vivo, and it demonstrated antinociceptive efficacy in mice after s.c. administration with reduced adverse effects of conventional opioids. Furthermore, LENART01 is the first peptide-based MOR-D2R ligand known to date. Our in vivo data indicate that LENART01’s activity as an MOR agonist may contribute to the beneficial effect of antinociception, while the dopamine D2R agonism may be favorable for the improved side effect profile as regards physical dependence. The current results, together with the recent report [46], reveal LENART01 as a ligand with MOR and D2R agonist activity. We also propose that the current and previously reported LENART01’s pharmacology might be supported by the molecular model findings [46], with its binding mode and interaction mechanisms with the two targets, the MOR and D2R. Further studies need to be conducted to establish LENART01’s efficacy in chronic pain models and provide additional data on its safety profile. A bivalent peptide-based drug design to engage both the MOR and dopamine D2R may represent a promising strategy in the pursuit of a novel class of opioid analgesics devoid of opioid liabilities.

## Figures and Tables

**Figure 1 ijms-25-04007-f001:**
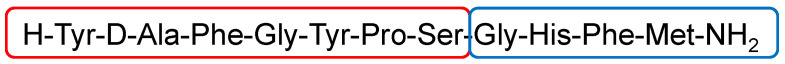
The amino acid sequence of LENART01. The dermorphin pharmacophore is framed in red, and the ranatensin analogue pharmacophore is framed in blue.

**Figure 2 ijms-25-04007-f002:**
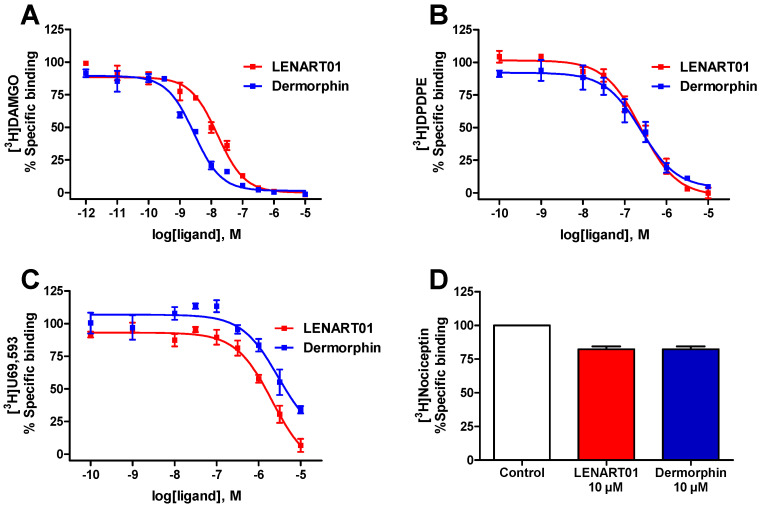
In vitro binding of LENART01 and dermorphin to human opioid receptors determined in radioligand competition binding assays. (**A**) Concentration-dependent inhibition of [^3^H]DAMGO binding to CHO-hMOR cell membranes. (**B**) Concentration-dependent inhibition of [^3^H]DPDPE binding to CHO-hDOR cell membranes. (**C**) Concentration-dependent inhibition of [^3^H]U69,593 binding to CHO-hKOR cell membranes. (**D**) Specific binding to human NOP receptor using [^3^H]nociceptin and CHO-hNOP cell membranes. Values represent means ± SEM (n = 3 independent experiments).

**Figure 3 ijms-25-04007-f003:**
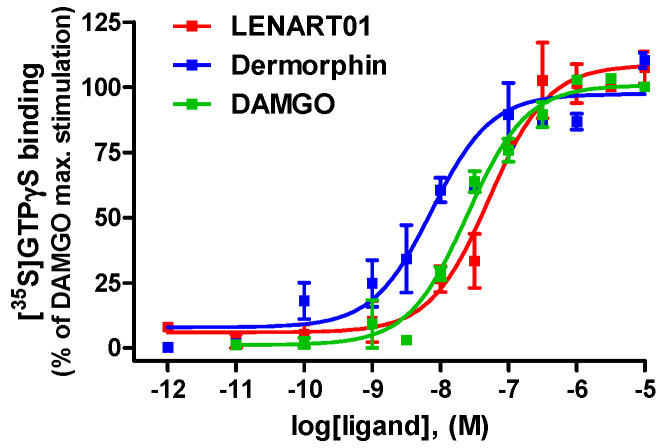
In vitro agonist activity of LENART01 and dermorphin to human MOR. Concentration-dependent stimulation of [^35^S]GTPγS binding by LENART01, dermorphin and DAMGO determined in the [^35^S]GTPγS binding assay using membranes from CHO cells expressing the human MOR. Percentage stimulation is presented relative to maximum simulation of reference MOR agonist, DAMGO. Values represent means ± SEM (n = 3–4 independent experiments).

**Figure 4 ijms-25-04007-f004:**
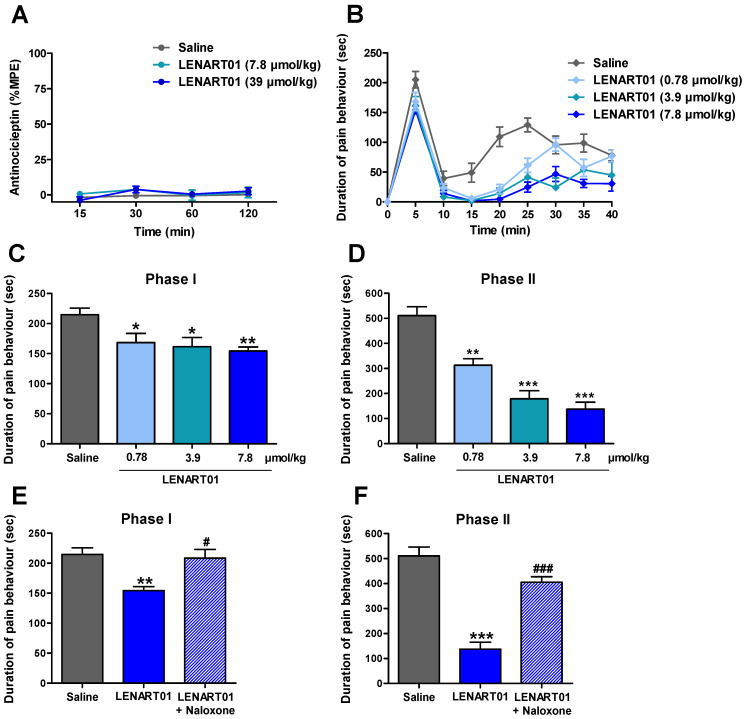
Effect of LENART01 in models of acute nociceptive pain and inflammatory pain after s.c. administration in mice. (**A**) Radiant heat tail-flick test. Groups of mice received s.c. saline or different doses of LENART01 (3.9 and 7.8 µmol/kg), and tail-flick latencies were measured before and after drug administration at different times. Antinociceptive response is expressed as % of maximum possible effect (%MPE). Values represent means ± SEM (n = 5 mice per group). (**B**–**F**) Formalin test. Groups of mice received s.c. saline (control) or different doses of LENART01 (0.78, 3.9 and 7.8 µmol/kg) before an intraplantar injection of the formalin solution in right hind paw. Duration of pain behavior was monitored for 40 min, determined as amount of time (in seconds, sec) each animal spent licking, biting, lifting and flinching formalin-injected paw. Time course of pain behavior in formalin test on mice treated with LENART01 (**B**). Dose-dependent antinociceptive effect of LENART01 during nociceptive Phase I (**C**) and inflammatory Phase II (**D**) of the formalin test. (**E**,**F**) Opioid antagonism by naloxone on antinociceptive effect of LENART01 during Phase I and Phase II. Mice were s.c. pre-treated with naloxone (1 mg/kg), 15 min before s.c. administration of LENART01 (7.8 µmol/kg). Values represent means ± SEM (n = 6–8 mice per group). * *p* < 0.05, ** *p* < 0.01, *** *p* < 0.001 vs. saline group; ^#^
*p* < 0.01, ^###^
*p* < 0.001 vs. LENART01-treated group; one-way ANOVA with Tukey’s *post hoc* test.

**Figure 5 ijms-25-04007-f005:**
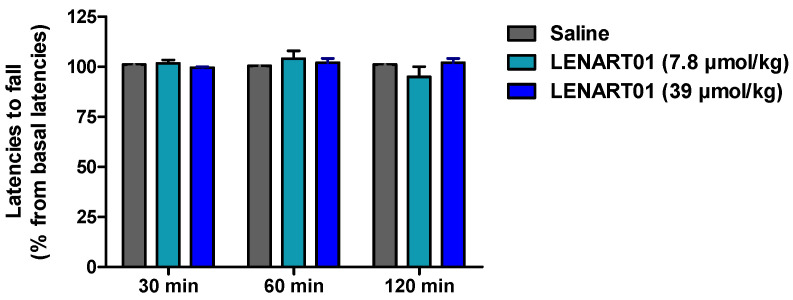
Effect of LENART01 in the rotarod test after s.c. administration in mice. Groups of mice received s.c. saline (control) or different doses of LENART01 (7.8 and 39 µmol/kg), and latencies to fall from the rotarod were measured before and after drug administration at different times. Latencies to fall are expressed as percentage (%) from baseline. Values represent means ± SEM (n = 5 mice per group).

**Figure 6 ijms-25-04007-f006:**
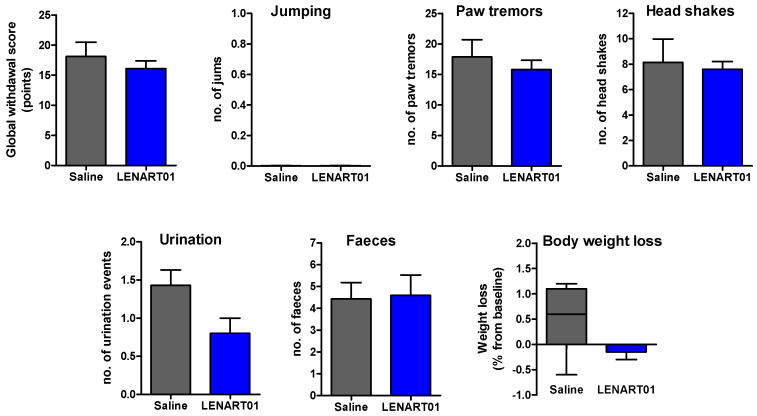
Withdrawal syndrome precipitated by naloxone after repeated s.c. treatment in mice with LENART01. Naloxone-induced withdrawal signs were assessed in mice that were treated twice daily for 5 days with saline (control) or LENART01 (7.8 µmol/kg). Withdrawal was precipitated two hours after last drug administration using naloxone (1 mg/kg, s.c.), and signs of withdrawal were counted over 15 min immediately after naloxone, and a global withdrawal score was calculated. Values represent means ± SEM (n = 5–7 mice per group).

**Figure 7 ijms-25-04007-f007:**
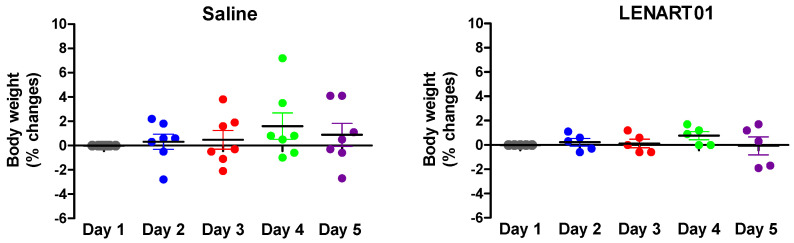
Weight changes in mice after chronic treatment with LENART01. Groups of mice received s.c. saline (control) or 7.8 mg/kg of LENART01 twice daily for 5 days, and changes in body weight were measured daily and are expressed as percentage (%) change from day 1. Values represent means ± SEM (n = 5–7 mice per group).

**Figure 8 ijms-25-04007-f008:**
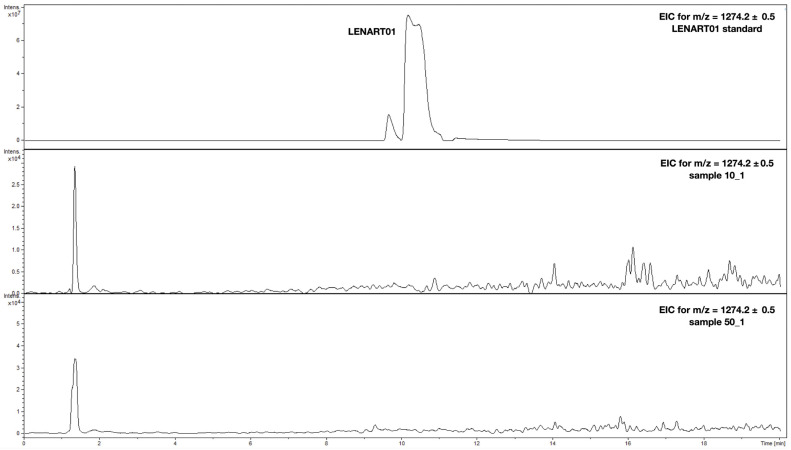
BBB permeability of LENART01. Extracted ion chromatograms (EICs) calculated for [M–H] = 1274.2 ± 0.5 *m*/*z* for standard hybrid peptide LENART01 (top) and brain homogenate samples from mice that were s.c. treated with 7.8 µmol/kg (middle) and 39 µmol/kg (bottom) of LENART01.

**Table 1 ijms-25-04007-t001:** In vitro binding affinities of LENART01 and dermorphin to human opioid receptors.

Ligand	Opioid Receptor Binding (K_i_, nM)				Selectivity K_i_ Ratios		
	MOR	DOR	KOR	NOP	DOR/MOR	KOR/MOR	NOP/MOR
LENART01	11.0 ± 1.5	227 ± 47	1306 ± 226	>10,000	21	119	>900
Dermorphin	1.89 ± 0.2	165 ± 23	2710 ± 873	>10,000	87	1434	>5000

The values were determined in radioligand competition binding assays using membranes from CHO cells stably expressing one of the human opioid receptors, MOR, DOR, KOR or NOP receptor. The values represent the means ± SEM (n = 3 independent experiments).

**Table 2 ijms-25-04007-t002:** In vitro agonist activities of LENART01 and dermorphin to human MOR.

Ligand	EC_50_ (nM)	% stim.
LENART01	48.5 ± 10.6	110 ± 7
Dermorphin	9.55 ± 2.58	104 ± 8
DAMGO	22.2 ± 4.7	100

Determined in [^35^S]GTPγS binding assay using membranes from CHO cells stably expressing human MOR. Percentage stimulation (% stim.) is relative to reference MOR agonist DAMGO. Values represent means ± SEM (n = 3–4 independent experiments).

**Table 3 ijms-25-04007-t003:** Antinociceptive effect of LENART01 in the formalin test after s.c. administration in mice.

Dose s.c.	Phase I %Antinociceptive Effect	Phase II %Antinociceptive Effect
0.78 µmol/kg	21.6 ± 7.2	38.8 ± 5.1
3.9 µmol/kg	24.8 ± 7.1	65.0 ± 6.4
7.8 µmol/kg	28.2 ± 3.1	73.1 ± 5.5
ED50 (µmol/kg) (95% CL)	- ^a^	1.60 (0.74–3.74)

Data are shown as percent (%) antinociceptive effect calculated for nociceptive pain (Phase I) and inflammatory pain (Phase II). Antinociceptive effective dose (ED_50_) and 95% confidence limit (CL) values were calculated using linear regression. - ^a^ not calculable. Values represent mean ± SEM (n = 6–8 mice per group).

## Data Availability

Data are contained within the article.

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
