# Peer review of "In Vitro and In Vivo Pharmacological Profiles of LENART01, a Dermorphin–Ranatensin Hybrid Peptide"

_ijms, 2024, doi:10.3390/ijms25074007_

Round 1

Reviewer 1 Report

Comments and Suggestions for Authors

Investigators studied in vitro and in vivo pharmacologic efficacy of the peptide-based hybrid compound, LENART01, that has activity on both MOR and dopamine D2 receptor. The compound is a combination of dermorphin (MOR agonist) and ranatensin (D2R agonist, bombesin-like peptide). The aim of combining D2R ligand with the MOR agonist was to prevent opioid-induced side effects like dependence. Investigators studied in vitro binding and functional activity of LENART01 on human mu, kappa, delta and nociceptin receptors using radioligand binding and GTPγS binding assays utilizing CHO cells that express receptors. They found that the LENART01 binds and activates human MORs. Antinociceptive activity of systemic administration of LENART01 was studied in two different pain assays in mice.  LENART01 was found inactive in tail flick assay but had a dose-related efficacy on second phase of formalin-induced inflammatory pain. The antinociceptive effect was abolished by pretreatment with naloxone suggesting the effect was through MOR. The hybrid compound did not induce sedation/motor incoordination. Additionally, chronic administration of the compound 2 times a day for 5 days did not induce physical dependence. Withdrawal symptoms (jumping, paw tremors, head shakes, increased defecation, weight loss) were similar in mice that received LENART01 to that of treated with saline chronically.  Lastly, they measured brain levels of the compound following systemic administration in mouse brain homogenates.  It was found that BBB penetrability of LENART01 is less. They concluded that because of less brain penetrability of the compound LENART01 was inactive at tail flick test since it is spinal.

Comments: It is a well-written study that indicates effectiveness of a hybrid compound with an analgesic effect without producing dependence. Throughout the manuscript, in some places ranatensin was written as ranhatensin. Additionally, the title of the 3.10 needs to be changed to one for pharmacokinetic studies. 

Just for a curiosity, since ranatensin is a bombesin-like peptide and those peptides are involved in itch sensation modulation, did researchers notify any scratching behavior following the administration of the compound?    

Author Response

Investigators studied in vitro and in vivo pharmacologic efficacy of the peptide-based hybrid compound, LENART01, that has activity on both MOR and dopamine D2 receptor. The compound is a combination of dermorphin (MOR agonist) and ranatensin (D2R agonist, bombesin-like peptide). The aim of combining D2R ligand with the MOR agonist was to prevent opioid-induced side effects like dependence. Investigators studied in vitro binding and functional activity of LENART01 on human mu, kappa, delta and nociceptin receptors using radioligand binding and GTPγS binding assays utilizing CHO cells that express receptors. They found that the LENART01 binds and activates human MORs. Antinociceptive activity of systemic administration of LENART01 was studied in two different pain assays in mice.  LENART01 was found inactive in tail flick assay but had a dose-related efficacy on second phase of formalin-induced inflammatory pain. The antinociceptive effect was abolished by pretreatment with naloxone suggesting the effect was through MOR. The hybrid compound did not induce sedation/motor incoordination. Additionally, chronic administration of the compound 2 times a day for 5 days did not induce physical dependence. Withdrawal symptoms (jumping, paw tremors, head shakes, increased defecation, weight loss) were similar in mice that received LENART01 to that of treated with saline chronically.  Lastly, they measured brain levels of the compound following systemic administration in mouse brain homogenates.  It was found that BBB penetrability of LENART01 is less. They concluded that because of less brain penetrability of the compound LENART01 was inactive at tail flick test since it is spinal.

Author’s reply: We would like to thank the Reviewer for reviewing our manuscript and the constructive comments. We have given careful consideration to all issues raised by you as indicated in the point-by-point response. Changes are highlighted in the revised manuscript.

Comments: It is a well-written study that indicates effectiveness of a hybrid compound with an analgesic effect without producing dependence. Throughout the manuscript, in some places ranatensin was written as ranhatensin. Additionally, the title of the 3.10 needs to be changed to one for pharmacokinetic studies. 

Author’s reply: We apologize for the errors. We have corrected the typo errors and the heading of section 3.10 (see Manuscript).

Just for a curiosity, since ranatensin is a bombesin-like peptide and those peptides are involved in itch sensation modulation, did researchers notify any scratching behavior following the administration of the compound?    

Author’s reply: We did not observe any scratching behavior induced by LENART01 in mice after s.c. administration at any of the tested doses in the pain assays, rotarod test and withdrawal syndrome assay.

Reviewer 2 Report

Comments and Suggestions for Authors

This is an interesting report of a hybrid molecule reported previously to have mu opioid receptor (MOR) and dopamine D2 receptor (D2R) activity. Most of the experiments provide nice insight as to the MOR related effects.

This manuscript adds very little to other MOR active peptides. There is value in a dual acting ligand but this was not shown. Only the MOR activity was profiled.

The authors need to add measures that address the dopaminergic effects in vivo.

The conclusions being drawn on MOR activity are valid. However, the conclusion that the effects are the result of dual action in vivo are not.

The major issue is there is no evidence of the D2 activity in vivo. If one is going to suggest a dual activity compound, dual activity should be shown.

Comments on the Quality of English Language

English is fine.

Author Response

This is an interesting report of a hybrid molecule reported previously to have mu opioid receptor (MOR) and dopamine D2 receptor (D2R) activity. Most of the experiments provide nice insight as to the MOR related effects.

Author’s reply: We would like to thank the Reviewer for reviewing our manuscript and the constructive comments. We have given careful consideration to all issues raised by you as indicated in the point-by-point response.

This manuscript adds very little to other MOR active peptides. There is value in a dual acting ligand but this was not shown. Only the MOR activity was profiled.

The authors need to add measures that address the dopaminergic effects in vivo.

The conclusions being drawn on MOR activity are valid. However, the conclusion that the effects are the result of dual action in vivo are not.

The major issue is there is no evidence of the D2 activity in vivo. If one is going to suggest a dual activity compound, dual activity should be shown.

Author’s reply: We do not share the view of the Reviewer that “This manuscript adds very little to other MOR active peptides”. The novel peptide LENART01 described in this manuscript and in our recent articles (Serafin et al. Molecules 2023, 28, 4955 and Serafin et al. Molecules 2024, 29, 272) was designed as a hybrid peptide with dual activity at the opioid and dopamine receptors. Furthermore, it is the first hybrid peptide reported with such profile. Only few small molecules as bifunctional opioid-dopamine receptor ligands were reported in the literature, specifically in two publications by Bonifazi et al. J. Med. Chem. 2021, 64, 7778-7808 and Bonifazi et al. J. Med. Chem. 2023, 66, 10304-10341. Whereres the two publications by Bonifazi et al. are solely SAR studies based on in vitro activies, our current study presents the in vivo pharmacology of the first dual opioid-dopamine ligand. The novelty of our study with LENART01 showing antinociceptive efficacy in mice after s.c. administration with reduced side effects has been noted throughout the manuscript (see Abstract, Results and Discussion and Conclusions).

To the comment on the D2 activity in vivo: In our study, we did not further evaluate the contribution of the D2 receptor in LENART01’s antinociception, given that pre-treatment of mice with the opioid antagonist naloxone resulted in a significant and complete reversal of the antinociceptive response of LENART01 after s.c. administration in the formalin test (see lines 247-249, Figure 4E and F). However, we show that LENART01’s agonist activity at the D2 receptor may have a beneficial effect in opioid-mediated liability of physical dependence, as LENART01 did no induce withdrawal syndrome after repeated administration in mice. The lack of inducing withdrawal syndrome by LENART01 is opposed to the effect of morphine, a MOR selective agonist. We have already stated this difference in the original submission (see lines 304-312).     

For better reading and understanding of our results, we have revised the Conclusions (lines 551-553): “While the MOR agonist activity of LENART01 contributes to the beneficial effect of antinociception, the dopamine D2R agonism appears to be favorable for the improved side effect profile as regards physical dependence”.

Round 2

Reviewer 2 Report

Comments and Suggestions for Authors

The authors do not provide enough evidence to conclude that the activity seen is the result of MOR agonism and D2 activity. The activity seen could be the result of different pharmacokinetic or pharmacodynamics. Additional studies are needed.

Author Response

The authors do not provide enough evidence to conclude that the activity seen is the result of MOR agonism and D2 activity. The activity seen could be the result of different pharmacokinetic or pharmacodynamics. Additional studies are needed.

Author’s reply: We thank the Reviewer for the additional review of our manuscript.

We respect the view of the Reviewer. However, in the previous round of revisions, we have discussed that our behavioral data provide evidence in the LENART01’s activity as the result of MOR agonism and D2R agonism. The MOR agonist activity contributes to the beneficial effect of antinociception (complete reversal of antinociception by the opioid antagonist naloxone, Figure 4E and F, lines 245-250). The dopamine D2R agonism accounts for the absence of inducing the unwanted side effect of physical dependence (Figure 6, lines 310-313). This was also summarized in the Conclusions. The profile of LENART01, as a MOR-dopamine D2 receptor hybrid ligand, is different to the effect of morphine, a MOR selective agonist, well-established to cause withdrawal syndrome by lowering the dopamine release and decreasing dopamine D2 receptor mRNA expression in the brain (published literature Acquas, E.; Eur. J. Pharmacol. 1991; Crippens, D. Brain Res. 1994; Georges, F. Eur. J. Neurosci. 1999, Fox, M.E.; Neuropsychopharmacology 2017).

Furthermore, the current study together with our previous report (Serafin et al. Molecules 2024, 29, 272) describe the pharmacokinetic and pharmacodynamics of LENART01 as a hybrid peptide, this includes also receptor binding and functional activity studies that revealed its agonistic character towards MOR and D2R. Additionally, our recent in silico studies (Serafin et al. Molecules 2024, 29, 272) using molecular docking and molecular dynamics simulations provided strong understandings on the binding mode and interaction mechanisms of LENART01 with the two targets, the MOR and D2R. Thus, the established experimental pharmacology of LENART01 is supported by the in silico findings. In our opinion, such results are the first to prove LENART01’s molecular targets.

Therefore, we consider that we provide clear evidence on the activity of LENART01 as a ligand with MOR and D2R activity. We have made additions in the Conclusions (lines 539-543).

While in this manuscript, we present first results on pharmacology of LENAR01, its antinociceptive efficacy and low-liability profile for sedation and withdrawal syndrome, our next goal is to further investigate efficacy of this hybrid peptide in animal models of chronic pain and additional behavioral data in its safety profile, particularly analgesic tolerance and addictive potential (tested in the CCP assay). We have made additions in the Conclusions (lines 543-544).

Round 3

Reviewer 2 Report

Comments and Suggestions for Authors

The authors have made a good faith attempt at addressing the concerns raised in the previous review. However, it would appear that there is a clear difference of opinion.

This reviewer feels the same issues remain. It is unclear if the effects seen are the result of D2 activity. In Serafin et al. Molecules 2024, 29, 272, the authors do not provide any information as to the compound is an agonist or antagonist. They only provide binding information. More enthusiasm would be for the peptide to produce a characteristic D2 agonist response. This reviewer is also not convinced that a computational study proves in vivo activity. The authors do provide some evidence that the peptide is unable to cross the BBB.  Unfortunately, there is not information on the half-life of the peptide in question. With these questions remaining, the authors' conclusions are tenuous.

Author Response

The authors have made a good faith attempt at addressing the concerns raised in the previous review. However, it would appear that there is a clear difference of opinion.

Author’s reply: We thank the Reviewer for the additional review of our manuscript. We appreciate the critical view of the Reviewer on our manuscript.

This reviewer feels the same issues remain. It is unclear if the effects seen are the result of D2 activity. In Serafin et al. Molecules 2024, 29, 272, the authors do not provide any information as to the compound is an agonist or antagonist. They only provide binding information. More enthusiasm would be for the peptide to produce a characteristic D2 agonist response. This reviewer is also not convinced that a computational study proves in vivo activity. The authors do provide some evidence that the peptide is unable to cross the BBB.  Unfortunately, there is not information on the half-life of the peptide in question. With these questions remaining, the authors' conclusions are tenuous.

Author’s reply: In our published study, Serafin et al. Molecules 2024, 29, 272, we have provided information that LENART01 is an agonist at the D2R in the [35S]GTPγS functional assay in both rat brain and spinal cord. The prove on the selectivity as a D2R agonist was based on experimental functional data as the selective D2R antagonist, risperidone, significantly reversed the stimulation of LENART01 to the basal activity. The selective D1R antagonist, SCH−39166, did not significantly reversed the stimulated [35S]GTPγS binding by LENART01. Therefore, we show experimental evidence on LENART01 as a D2R agonist.

We agree with the Reviewer that computational data cannot prove the in vivo activity. However, the combination of in silico and experimental evaluation is nowadays an important approach in understanding the biological activity of (know and new) ligands. In our previous study, Serafin et al. Molecules 2024, 29, 272, the computational work provided first time information on the binding mode and interaction mechanisms of LENART01 with the two targets, the MOR and D2R. We only propose that our current and reported pharmacology of LENART01 might be supported by the molecular modeling data. 

We have amended the Conclusions.

Round 4

Reviewer 2 Report

Comments and Suggestions for Authors

The authors have attempted to address the concerns raised previously.